# No Excess Mortality up to 10 Years in Early Stages of Breast Cancer in Women Adherent to Oral Endocrine Therapy: A Probabilistic Graphical Modeling Approach

**DOI:** 10.3390/ijerph19063605

**Published:** 2022-03-18

**Authors:** Ramon Clèries, Maria Buxó, Mireia Vilardell, Alberto Ameijide, José Miguel Martínez, Rebeca Font, Rafael Marcos-Gragera, Montse Puigdemont, Gemma Viñas, Marià Carulla, Josep Alfons Espinàs, Jaume Galceran, Ángel Izquierdo, Josep Maria Borràs

**Affiliations:** 1Hospitalet de Llobregat Avenue Gran Vía 199-203, 08908 Barcelona, Spain; rfont@iconcologia.net (R.F.); ja.espinas@iconcologia.net (J.A.E.); jmborras@iconcologia.net (J.M.B.); 2Institut d’Investigació Biomèdica de Bellvitge (IDIBELL), Hospitalet de Llobregat, Avenue Gran Via de l’Hospitalet, 199-203-1a Planta, 08908 Barcelona, Spain; 3Department de Ciències Clíniques, Universitat de Barcelona, 08907 Barcelona, Spain; 4Institut d’Investigació Biomèdica de Girona (IDIBGI), C/Dr. Castany s/n, Edifici M2, Parc Hospitalari Martí i Julià, 17190 Girona, Spain; mbuxo@idibgi.org (M.B.); rmarcos@iconcologia.net (R.M.-G.); 5Independent Researcher, 08700 Barcelona, Spain; mvilardelln@uoc.edu; 6Registre de Càncer de Tarragona, Servei d’Epidemiologia i Prevenció del Càncer, Hospital Universitari Sant Joan de Reus (IISPV), 43204 Reus, Spain; alberto.ameijide@salutsantjoan.cat (A.A.); maria.carulla@salutsantjoan.cat (M.C.); jaume.galceran@salutsantjoan.cat (J.G.); 7Department de Estadística i Investigació Operativa, Universitat Politècnica de Catalunya (EDIFICI H), Diagonal 647, 08028 Barcelona, Spain; jose.miguel.martinez-martinez@upc.edu; 8Grupo de Investigación en Salud Pública, Universidad de Alicante, 03690 Alicante, Spain; 9Registre de Cáncer de Girona-Unitat d’Epidemiologia, Institut Català d’Oncología, 17005 Girona, Spain; mpuigdemont@iconcologia.net (M.P.); aizquierdo@iconcologia.net (Á.I.); 10Grup d’Epidemiologia Descriptiva, Genètica i Prevenció del Càncer de Girona-IDIBGI, 17005 Girona, Spain; 11Facultat de Medicina, Universitat de Girona (UdG), 17071 Girona, Spain; 12Centro de Investigación Biomédica en Red, Epidemiología y Salud Pública (CIBERESP), 28029 Madrid, Spain; 13Servei d’Oncología Médica, Institut Català d’Oncología, Hospital Universitari de Girona “Doctor Josep Trueta”, 17005 Girona, Spain; Gvinyes@iconcologia.net

**Keywords:** breast cancer, excess mortality, adherence, endocrine therapy, synthetic dataset, graphical modeling

## Abstract

Breast cancer (BC) is globally the most frequent cancer in women. Adherence to endocrine therapy (ET) in hormone-receptor-positive BC patients is active and voluntary for the first five years after diagnosis. This study examines the impact of adherence to ET on 10-year excess mortality (EM) in patients diagnosed with Stages I to III BC (N = 2297). Since sample size is an issue for estimating age- and stage-specific survival indicators, we developed a method, ComSynSurData, for generating a large synthetic dataset (SynD) through probabilistic graphical modeling of the original cohort. We derived population-based survival indicators using a Bayesian relative survival model fitted to the SynD. Our modeling showed that hormone-receptor-positive BC patients diagnosed beyond 49 years of age at Stage I or beyond 59 years at Stage II do not have 10-year EM if they follow the prescribed ET regimen. This result calls for developing interventions to promote adherence to ET in patients with hormone receptor-positive BC and in turn improving cancer survival. The presented methodology here demonstrates the potential use of probabilistic graphical modeling for generating reliable synthetic datasets for validating population-based survival indicators when sample size is an issue.

## 1. Introduction

Breast cancer (BC) is the most common cancer and the leading cause of cancer death in European women [1]. A decrease in BC mortality is correlated with improvements in survival [2,3], an indicator of the success of cancer control efforts in a population-based setting. Conditional five-year survival is an outcome that measures the efficacy of cancer management, since it responds to the question of “once a patient survives for T years, what is the probability of surviving another five years?” [4]. Most population-based cancer survival indicators are derived from relative survival (RS), defined as the ratio between the overall survival (OS) and expected survival of the cohort with respect to the general population [5]. RS is as an estimate of the patients’ cancer-specific survival compared to the survival of the general population, and one can also assess the conditional RS(CRS) at five additional years after surviving T years [5]. On the basis of the CRS(T), one can determine the five-year excess mortality as EM(T) = 1-CRS(T), which is used to assess whether patient mortality surpasses the mortality of the general population, that is, when EM(T) > 0 [6].

These conditional survival or mortality indicators provide very relevant information on the prognosis of BC over time, as they are a starting point to identify prognostic factors related to long-term survival [6,7,8,9]. For instance, the BC cohort’s mortality is not different from the general population’s mortality when EM equals 0 beyond a certain time interval T [6]. Moreover, population-based cancer registries can define the time to cure of cancer as “the number of years after cancer diagnosis when the EM, expressed as a percentage, becomes negligible” [4,8]. That situation occurs when the EM remains clearly below 5% for more than 10 years, and CRS consequently surpasses 95% [8]. A recent study using European cancer registry data showed that an EM of 5% could persist for at least 15 years in BC patients [9]. However, the EM in that study was an overall indicator that could only be adjusted for age because other prognostic factors could not be retrieved from all participating cancer registries.

Stage, molecular subtype, and adherence to endocrine therapy (ET) are key predictors for providing population-based BC survival estimates [10]. Indeed, tamoxifen and aromatase inhibitors are pillars of adjuvant therapy for patients with hormone receptor positive (HR+) BC diagnosed at Stages I–III [11]. Randomized clinical trials showed that five years of adherence to ET positively impact BC survival [11]. In a previous study, we found that nonadherence to ET is significantly and independently associated with recurrence and all-cause mortality at Stages I–III of hormone receptor positive BC after adjusting for age [12]. A question arises regarding the impact of ET adherence on long-term survival and risk of death in patients with BC versus the general population [9].

The sample size of the cohort could be an issue when trying to estimate age-specific survival according to stage and molecular subtype; however, generating a large cohort of simulated survival data on the basis of observed cohort data could help overcome this limitation [13]. This simulation could be achieved in two ways: (1) only simulating survival times [14,15] or (2) generating a set of cohort covariates as a function of survival times [13,16]. For the latter, oversampling techniques such as SMOTE [17], Borderline SMOTE [18], and MWMOTE [19] can also be used to generate balanced subsets of data, where the efficiency of these methods in simulating new datasets must be assessed with the observed survival patterns of real data [19]. However, if we are interested in detecting new patterns of survival, the specific modeling of probabilistic dependencies between the variables of the observed data is needed, which requires estimating a joint probability distribution of the variables [13,20,21,22]. For that purpose, our research team developed Modelling Graphical Probabilistic Dependencies (ModGraProDep) and suggested that future work should be oriented toward selecting data subsets across several synthetic datasets (SynD) that better mimic the cohort’s survival pattern [13].

In the present study, we developed a method to validate the survival estimates of the original cohort by using a synthetic cohort that combines the “best” subsets of simulated data derived from graphical models. Survival indicators are generated by fitting the cohort data and the simulated SynD to a Bayesian RS model developed for that purpose.

## 2. Materials and Methods

### 2.1. Data: BCStage Dataset

BC data were obtained from the population-based cancer registries of Girona and Tarragona (northeastern Spain) covering an average annual population of 560,120 women from 2005 to 2009 [23]. During this time period, 4053 women under the age of 75 years were diagnosed with invasive BC (code C50 of the 10th edition of the International Classification of Diseases, ICD-10). A total of 352 women (8.7%) were excluded from the analyses due to missing data on estrogen and progesterone status, and another 1215 (30.0%) were excluded due to missing data on stage, Stage IV at diagnosis, or diagnosis of HER2-enriched or triple-negative BC tumors, and we could not retrieve follow-up status (if the patient died or not at the end of follow-up) in N = 189. Each woman with BC diagnosed from 2005 to 2009 was followed up to 31 December 2019; we considered a maximal follow-up of 10 years. Of the patients eligible for ET (N = 2297), information could only be retrieved for BC patients diagnosed from 2007 to 2009 who met the inclusion criteria: patients presenting positivity for estrogen and/or progesterone receptors diagnosed at Stages I, II, or III, who were eligible for ET (N = 1243). Survival times for patients not found to be dead at the end of follow-up were censored. Stage classification was based on the TNM classification system, as described in the 6th edition of the American Joint Committee on Cancer staging manual [24], classifying patients at Stage I, II, or III when TNM was available at the moment of diagnosis.

Adherence to ET for patients with HR+ BCs was tracked during the first five years after BC diagnosis. Any switch to tamoxifen or aromatase inhibitor was considered to be a continuation of treatment. Adherence was estimated as “the proportion of days covered by a filled drug prescription over the treatment period (up to five years from the date of first prescription)”, deeming a cumulative adherence rate of 80% or more as satisfactory [12]. Data on ET prescription refills for BC were collected for the entire study period (2007–2015) from the community pharmacy database, which is mandatory for drug reimbursement in Catalonia.

Collected variables were: age (26, 27, …, 73, 74), stage at diagnosis (I, II, or III), adherence to ET (yes: adherence rate > 80% vs. no: adherence rate ≤ 80%), follow-up years (1, …, 10) and exitus (died vs. survived). Age was also considered to be a categorical variable with three age groups: ≤49, 50–59, and 60–74 years. Patients were additionally classified according to the tumor positivity of the human epidermal growth factor receptor (HER2) expression (HER2+ vs. HER2−).

### 2.2. Synthetic Data Simulation

#### 2.2.1. Fitting Graphical Models through ModGraProDep

Four synthetic datasets were simulated by modeling the probabilistic dependencies between variables using ModGraProDep [13]. In brief, let Γ be the set of cells in a contingency table, where $c_{ash}$ is a cell of the table with indices a(age)−s(stage)−h(adherence). Let $p\left(c_{ash}\right)$ be the cell probabilities of the contingency table Γ. Using a hierarchical expansion of log ($p\left(c_{ash}\right)$) we considered a saturated log-linear model, a model including the main effects, and all interactions between these, that is
(1)$$\log\left(p(C_{ash}\right))=\alpha +\beta_{a}+\beta_{s}+\beta_{h}+\gamma_{2}^{I}+\gamma_{3}^{I}$$
where parameter α is an intercept, β  refers to main effects, and $\;\gamma_{q}^{I}\;$ refers to the set of interaction parameters of order *q*, where  q∈{2,3}. We can also specify a model with fewer interaction terms by setting higher-order interaction to zero.

Assuming that there is a set of candidate models  M(j)|j∈{1,…,J}, ModGraProDep uses a heuristic search based on penalized log-likelihood
*H(j,k) = −2log(p(c_ash_)) + k**∗z(j)*(2)
where  z(j) is the number of model parameters, and  k is a penalty factor. Changing the value of *k* can result in several models using backward stepwise elimination of graph arches. Starting from the saturated model, ModGraProDep fits four models: three by using the *k* penalty factor, GMK1 for *k* = 1, GMAIC for *k* = 2 (Akaike information criterion [25]), and GMBIC for *k* = log (N) (Bayesian information criterion [25]), and another by testing the arch’s conditional independence, GMTEST. Once these four models had been fitted, we first imputed adherence in the BC cases with missing adherence, and then generated the synthetic datasets. We used the junction-tree simulation algorithm implemented in ModGraProDep for simulating four datasets of size N = 1,000,000 from each of the four models and according to the probabilistic relationships between variables (see Vilardell et al. (2020) for technical details [13]).

#### 2.2.2. ComSynSurData: Combining Synthetic Survival Datasets 

Figure 1 presents the scheme for generating a combined synthetic dataset that selects the best subsets of data that better mimic the survival pattern of the cohort. These are summarized as follows:
Step 0.Use ModGraProDep for generating the four SynDs.Step 1.Produce a partition of the cohort dataset into L subsets according to A age groups and S levels of a stratification variable, such as stage at diagnosis; then, L = A × S. For instance, if strata were stage at diagnosis with levels {I, II, III}, and three age groups were considered, then L = 3 × 3 = 9 subsets (one for each age group and stage combination). In the same line, the same partition is made for each SynD.Step 2.For each of the L subsets of the cohort data, find its “best” counterpart among the 4 × 9 = 36 subsets of SynDs by comparing survival estimates between the observed cohort and that derived from the SynDs through a scoring method.Step 3.Once L subsets of SynD are selected in each age stratum, generate a combined synthetic cohort by merging these L subsets, from which Kaplan–Meier survival estimates according to stage and corresponding age groups can be derived.

#### 2.2.3. Scoring Method for Comparing Observed versus Predicted Survival in Step 2

ComSynSurData uses the integrated Brier score (IBS), a scoring method to detect inaccuracies in the prognostic classification scheme, that is, disagreement between the survival curves of cohort and simulated data at a certain time *T* [26]. Let $\hat{S(}T)$ be the predicted survival function, and $\hat{G(}T)$ the censoring distribution, both functions estimated using the Kaplan–Meier method and using the SynD. Here, we used the following definition of the Brier score at time *T* for censored data [27]:(3)$$BS^{C}\left(t\right)=\frac{1}{n}\left[\overset{n}{\underset{i=1}{\sum }}\left[\frac{{\left(0-\hat{S\left(t\right)}\right)}^{2}}{\hat{G\left(t\right)}}I\left(t_{i}\leq t\right)+\frac{{\left(1-\hat{S\left(t\right)}\right)}^{2}}{\hat{G\left(t\right)}}I\left(t_{i}>t\right)\right]\right]$$
where $t_{i}$ is the follow-up of the *i*-th patient in the cohort, and *I*(·) are indicator functions, such that $I\left(t_{i}\leq t\right)\;$= 1 and $I(t_{i}>t)$ = 0 if the *i*-th patient dies before *t*, and $I(t_{i}>t)$ = 1 and $I\left(t_{i}\leq t\right)$ = 0 if the *i*-th patient does not die before *t*.

IBS is an overall measure up to a certain time target $t^{*},\;$which uses weights defined as W(*t*) = *t*/*t** [27]. Here, we used maximal follow-up $t^{*}$ = 10 years. The IBS was calculated as
(4)$$IBS^{C}\left(t^{*}\right)=\int_{0}^{t^{*}}BS^{C}\left(u\right)dW\left(u\right)=\frac{1}{10}\int_{0}^{t^{*}}BS^{C}\left(t\right)dt$$

For each age and stage stratum, the selected subset of *SynD* would be that with the smallest IBS score, which could lie between 0 and 1, where IB*S* = 0 shows a perfect match between observed and predicted survival [26,27]. The Appendix A includes the R code for running *ComSynSurData*.

### 2.3. Statistical Modeling of Excess Mortality

We used an RS model to derive the survival indicators. Let $\lambda_{O}\left(T\right)$ be the overall hazard of death in the cohort at a specific time *T*, and $\lambda_{P}\left(T\right)$ is the expected hazard in the cohort using the general population mortality [28]. Applying additive modeling, the excess hazard of death in the cohort due to BC is $\lambda_{X}\left(T\right)=\lambda_{O}\left(T\right)$ −$\;\lambda_{P}\left(T\right)$ [29], where *OS(T)* = $\int_{0}^{T}\exp\left(-\lambda_{O}\left(T\right)dt\right)$ is the observed survival in the cohort at time *T*, and *ES(T)* its expected survival in the cohort, *ES(T)* =$\;\int_{0}^{T}\exp\left(-\lambda_{P}\left(T\right)dt\right)$. Relative survival (RS) at time *t* is calculated as [28]:(5)$$RS\left(T\right)=\frac{OS\left(T\right)}{ES\left(T\right)}$$

*RS(T)* could reach (or even surpass) 1 when *OS(T)* is equal to the survival of the general population [28]. From *RS(T)*, one can derive the five-year conditional relative survival at *T* years of follow-up as [5]
(6)$$CRS\left(T\right)=\frac{RS\left(T+5\right)}{RS\left(T\right)}$$
From this, the five-year conditional excess mortality (EM) at *T* years of follow-up [5,6].
EM(T) = 1 − CRS(T)(7)

Using (7), one can assess temporal changes in the *EM* by monitoring this quantity during follow-up [5]. Moreover, it is of interest for both the patient and clinician to estimate the probability of death due to cancer in the presence of other causes at time *T*, PCa(*T*) and the crude probability of death due to other causes in the presence of cancer mortality at time *T*, *POC(T)* [6]. These quantities can be derived from the *RS(T)* by using competing risks modeling as
(8)$$PCa\left(T\right)=\int_{0}^{T}\mathrm{OS}\left(u\right)\lambda_{X}\left(u\right)du\;$$
(9)$$POC\left(T\right)=\int_{0}^{T}\mathrm{OS}\left(u\right)\lambda_{P}\left(u\right)du\;$$
where the sum of these two probabilities gives the probability of death from any cause at time *T* [6]. Since all these indicators are related to $\lambda_{X}\left(t\right)$ and $\lambda_{O}\left(t\right)$, these last two risks can be estimated by $\hat{\lambda_{O}}\left(T\right)=O\left(T\right)/Y\left(T\right)$ and $\hat{\lambda_{P}}\left(T\right)=E\left(T\right)/Y\left(T\right),$ where O(T) is the observed number of deaths at *T* and *E*(T) is the expected number of deaths at *T*, which is calculated from applying the age-specific mortality rates of the general population to each one of the individuals at risk within the *T* interval, and finally, *Y(T)* is the number of individuals at risk in *T*.

Since *O(T)* is usually considered to be a Poisson-distributed random variable with mean $\mu_{T}$, we used a Bayesian autoregressive modeling of order 1 to estimate $\hat{\lambda_{O}}$, assuming a prior precision (inverse of variance) of 0.001 [30], defined as
(10)$$O\left(T\right)\sim Poisson\left(\mu_{T}\right)\;log\left(\mu_{1}\right)=\log\left(Y\left(1\right)\right)+\delta_{1}\;\;\delta_{1}\sim N\left(0,0.001\right)\;$$
$$log\left(\mu_{T}\right)=\log\left(Y\left(T\right)\right)+\delta_{T}|T>1\;\delta_{T}\sim N\left(\delta_{T-1},0.001\right)|T>1$$

Posterior distributions and the corresponding 95% credible intervals of aforementioned survival indicators (5)–(9) were calculated through posterior estimates of $\mu_{T}$, and fixed quantities *E(T)* and *Y(T).* The model was implemented using WinBUGS [31] (see the program code in Appendix A), which was run within R (http://www.R-project.org, accessed on 5 December 2021) through the R2WinBUGS library [32].

### 2.4. Analysis Scheme

First, the GM was fitted to the original dataset, and adherence was imputed in cases with missing information. Second, four SynDs were generated using ModGraProDep, and from these SynDs, ComSynSurData selected the best L age-stage subsets of synthetic data that were used to generate the combined synthetic dataset. Survival indicators were derived from fitting the Bayesian relative survival model to this combined cohort, and these were also validated with those obtained using the original cohort. Lastly, age-specific survival indicators for epidemiologic or clinical use were calculated.

## 3. Results

Table 1 presents the clinical and pathological characteristics of the observed cohort in Girona and Tarragona in 2005–2009, stratified according to HER2+/HER2− expression. Main differences were detected in the distribution of BC stage: stages II and III were more frequent in patients with HER2+ compared to HER2− tumors. Mean age at diagnosis was 55.3 years: 32.7% of the patients were diagnosed with BC before 50 years of age, 29.6% were diagnosed at age 50 to 59 years, and 37.7% were 60 years or older. Most patients were diagnosed at early stages, whereas only 17.5% were diagnosed at Stage III. Mean follow-up was 8.2 years, and 11.7% of patients died during that period. Of these, information about adherence could be retrieved in those diagnosed from 2007 to 2009 (N = 1243), 75% of whom showed a cumulative adherence rate of 80% or higher during the first five years after the BC diagnosis. In cases with missing adherence data, a value for adherence was imputed making use of ModGraProDep.

Table 1 also shows the distribution of the number of BC cases according to adherence and HER2 status after the imputation of these four models. We did not find any difference in the distribution of the percentages according to adherence status when comparing the observed frequencies in the cohort (the N = 1243 BC patients) with those obtained after using each of the four models implemented in ModGraProDep (see Table 1, *Distribution of BC cases in the cohort after the imputation of adherence to ET when missing)*. However, the distribution of adherence status in the cohort was identical when GMAIC and GMBIC models were used, indicating that the probabilistic graphical pattern of the dependencies between variables in the observed data (N = 1243) was likely to be identical when fitting these two graphical models to the cohort data.

Figure 2 shows the graphical modeling of the data, which encodes a factorization of the joint probability distribution of the dataset. Three probabilistic schemes can be distinguished: one obtained using GMK1 (Figure 2a), another using GMTEST (Figure 2b), and another, as noted above, obtained through GMAIC and GMBIC (Figure 2c). Figure 2a shows that the model GMK1 considered that all variables were related (connected). The GMTEST model considers age as related to exitus, but this is conditional on adherence or stage at BC diagnosis, and HER2 as directly related to the other variables through stage. Lastly, GMAIC and GMBIC models consider that age could be independent from the data structure, and all remaining variables are conditionally independent once exitus is known. Stage was related to the remaining variables, conditional on others, regardless of the model used.

### 3.1. Data Simulation

After the imputation of the missing data, ModGraProDep was used for simulating the four SynDs, and from these, ComSynSurData was applied to generate the combined dataset. Four datasets were considered, and on their basis, four SynDs were simulated. Once these models were fitted, the four SynDs were introduced into ComSynSurData, and the combined dataset was generated. Table 2 shows the matrix of internally generated IBS scores by ComSynSurData from which to select the L = 9 subsets. From these, seven data subsets were selected from the SynD dataset derived from GMK1, two from the SynDs derived from GMAIC and GMBIC, and none from GMTEST.

### 3.2. Comparing Observed Survival in the Cohort with Survival in the Combined Cohort

To assess the reliability of these simulated datasets, OS in the cohort with real data (N = 1243) was compared with the estimated survival using the combined cohort (Figure 3). Using the posterior distribution of the survival derived from the combined cohort, its median survival overlapped with the 95% credible intervals of observed survival in the original cohort in almost all age groups. In some, however, the median of the survival’s combined cohort was slightly lower than the observed survival, but close to the lower bound of the 95% credible interval of the survival in the original cohort: age group ≤ 49 years at Stages I and II, and for the age group of 59–74 years at Stage III.

### 3.3. Survival Indicators Derived from Combined Dataset

Figure 4 compares the EM observed in the original cohort with that estimated using the combined dataset. Median EM between these datasets did not differ, since the 95% credible intervals derived from the observed cohort overlapped with the estimates derived from the combined cohort. In this line, the patients diagnosed in stages I and II who were adherent to endocrine therapy did not show EM with respect to the general population. However, we found that patients diagnosed in these early stages who were not adherent to ET had an EM with a median ranging from 5% to 10%, which usually suggests a significant EM. For patients diagnosed at Stage III, the effect of nonadherence to ET might double the EM with respect to adherence.

Table 3 presents the age-specific epidemiological survival indicators derived from the combined cohort across age groups and stage at diagnosis. The adherence group showed higher OS (+ 6% at 5 years and +15.2% at 10 years) and lower 10-year PCa (−18.7%) and 5-year EM (−14.5%) compared to the nonadherent group.

Table 3 also shows that, at Stage I, adherent patients diagnosed before 50 years of age may present a small but non-negligible 1.1% EM when compared to the general population. In contrast, no EM was detected in patients diagnosed beyond that age. Nonadherent patients present 4.6% to 9% higher EM, depending on the age group. I Stage II, adherent patients diagnosed beyond 59 years did not show EM during the follow-up. The largest differences in survival indicators between adherent and nonadherent patients were observed in the Stage III group, with better prospects of survival in adherent compared to nonadherent patients, independently of age at BC diagnosis.

Figure 5 shows the comparison of the 3 main population-based survival indicators across age groups and stratified by adherent and nonadherent patients: EM(5), PCa(10) and OS(10). In Stages I and II of BC, differences in EM(5) and PCa(10) between adherent and nonadherent patients were clearly marked and showed their maximum among BC patients diagnosed beyond 50 years. At Stage III, the age trend of these two indicators was similar, showing a marked rise beyond 59 years of age at BC diagnosis. Lastly, OS(10) showed two patterns: (i) for adherent patients, survival was similar up to 59 years of age and decrease thereafter, independently of stage at diagnosis; (ii) for nonadherent patients, OS(10) exponentially decreased with age except in Stage III.

## 4. Discussion

This study provides estimates of the most common population-based statistical indicators in order to assess the impact of stage, age, and adherence to ET for survival in patients with positive estrogen- and/or progesterone-receptor BC. We compared the estimates from the original cohort with those derived from synthetic datasets generated through graphical models fitted to the cancer registry cohort. Using the advantages of probabilistic graphical modeling, we first identified the probabilistic data structure, used it to impute the adherence status in patients with missing data for this variable, and simulated data for a large cohort to estimate age-specific survival indicators. We implemented the CombSynSurData method in order to select the best subsets of four synthetic datasets derived from ModGraProDep. To the best of our knowledge, this is the first study to show that adherence to ET greatly impacts BC survival among HR+ patients with early-stage breast cancer: no excess risk of death up to 10 years after BC in women diagnosed beyond 49 years of age. This result sheds light into curing BC for this group of patients.

The assessment of treatment response is crucial for evaluating anticancer therapies, treatment planning, and outcomes, where patients’ OS is the baseline measure [33]. However, that evaluation requires a large sample and long-term follow-up, which are usually not available in the same study. We used a method for generating a large sample of synthetic data on the basis of the original cohort in order to estimate the observed survival indicators using the original cohort data, which had the minimal required long-term follow-up of 10 years for assessing EM due to BC [8]. Using these indicators, healthcare policy planning should be informed by the estimated prevalence of cancer deaths at a population level, which can be calculated through RS [34]. These indicators are strongly related to the concept of a statistical assessment of the “cure” of BC [35], which entails: (I) long survival time beyond 10 years and equal life expectancy [9], and (II) no cancer relapses up to almost 10 years after BC diagnosis [35].

Our study has a strong limitation in assessing the statistical cure of BC: our follow- up cannot go beyond 10 years. Another limitation is that the simulated cohorts were based on the observed data provided by the original cohort. Therefore, survival indicators derived from these simulated cohorts can only internally validate the indicators estimated from the original data. The availability of external data provided by other cancer registries with similar information would be useful for an additional validation of the results and reproducibility. Information on long-term prognosis by stage, receptor status and adherence to ET is information not usually reported by population-based cancer registries [9]. However, recent studies suggest the need for using these variables for population-based studies in order to assess whether the influence of stage or BC subtype on survival lessens in the long term, which might lead to a consideration of cancer cure in early stages [36,37,38]. The impact of ET adherence on BC patient survival is significant [39], and our results, which show differences in EM when comparing the cohort’s mortality with that of the general population, are relevant to this. Moreover, differences between adherent and nonadherent patients are significant across all age groups, but show different impact depending on stage at diagnosis. This point must be accounted and further investigated, since age, stage, and treatment play a crucial role in the clinical follow-up of BC patient. Studies regarding this are needed.

A small but significant level of EM was detected in the adherent group of younger BC patients (<50 years) diagnosed at Stage I. However, survival estimates for these women using the combined cohort could be slightly lower than the observed survival in the original cohort, and this could limit the use of this subset of data. On the other hand, a previous study carried out on a cohort with ductal carcinoma in situ and diagnosed in Girona also detected statistically significant EM in patients diagnosed before 50 years of age [40]. Evidence suggests that differences in biological characteristics of breast tumors could impact patient survival [41]. Moreover, 5- and 10-year local recurrences at early stages [42] arise depending on age and molecular subtype. Although a high proportion of BCs are HR+ and HER2−, those diagnosed in young women are likely to be more aggressive [43,44], even in luminal-like early BC [45,46]. A study carried out using SEER data noted worse BC-specific survival for women in the oldest age groups for every BC subtype analyzed, with the exception of Stage IV triple-negative disease [10]. In that study and others, worse survival was observed in patients diagnosed before 35 years of age at Stages I–III [10,46]. Other studies showed that young age is also a predictor of decreased adherence to adjuvant ET, which in turn is associated with increased mortality [47]. Although ET is unquestionably a therapeutic tool for HR + BC, these strategies are associated with potential side effects and toxicity, which may have a differential effect depending on age [48]. On the other hand, randomized trials showed that, in premenopausal women with BC, the addition of ovarian suppression to tamoxifen may increase 8-year rates of both disease-free and overall survival [49]. However, diagnoses in the cohort under study predate results of these randomized trials, so women under 49 years of age in our study could not have had access to these improved treatments. Studies on BC survival and late adverse events due to ET must be considered beyond 10 years of follow-up, since evidence suggests that distant recurrences may arise from 5 to 20 years after diagnosis [49].

Studies of EM derived from small cohorts of cancer patients must be further evaluated using larger cohorts [40]. Here, we present a procedure for simulating a large sample dataset by fitting graphical models to cohort data and coupling a log-linear model and a Bayesian network. Since our interest was in simulating the most reliable data, one aim was to assess the probabilistic dependencies between variables. ModGraProDep identifies a set of graphical models by using a heuristic search based on changing *k*, a penalty factor in the partial likelihood (see Equation (2) above) [16,17,18,21]. Although specific values of *k* such as *k =* 2 and *log (N)* equation lead to two known measures for model choice, AIC and BIC, ModGraProDep identifies two alternative models, one using *k* = 1 and another testing the arch’s statistical significance at α = 0.05 [13,18]. Vilardell et al. showed that estimating survival from one of these four models could provide reliable survival indicators [13]. Here, we introduced a method for deriving a synthetic dataset that provides better survival indicators by combining the best subsets of data of several synthetic datasets. An interesting feature in ComSynSurData is that it could be adapted to use any set of simulated data, and these could come from oversampling techniques, such as SMOTE [17], Borderline SMOTE [18] and MWMOTE [19]. However, synthetic datasets derived from ModGraProDep provide additional information about the data structure and data relationship between variables. The latter can also be useful for clinicians and epidemiologists in understanding the probabilistic patterns of the disease under study.

## 5. Conclusions

To sum up, coupling relative survival modeling with synthetic data simulation validated our main clinical result: patients with HR+ breast cancers diagnosed beyond 49 years of age at Stage I and diagnosed beyond 59 years of age in Stage II do not have 10-year EM compared to the general population if they follow the prescribed regimen of ET. These results call for developing interventions that promote adjuvant ET adherence in eligible BC patients given its potential benefits in improving cancer survival. The methodology presented here demonstrates the potential use of probabilistic graphical modeling in generating reliable synthetic datasets to be used for validating population-based survival indicators when sample size is an issue.

## Figures and Tables

**Figure 1 ijerph-19-03605-f001:**
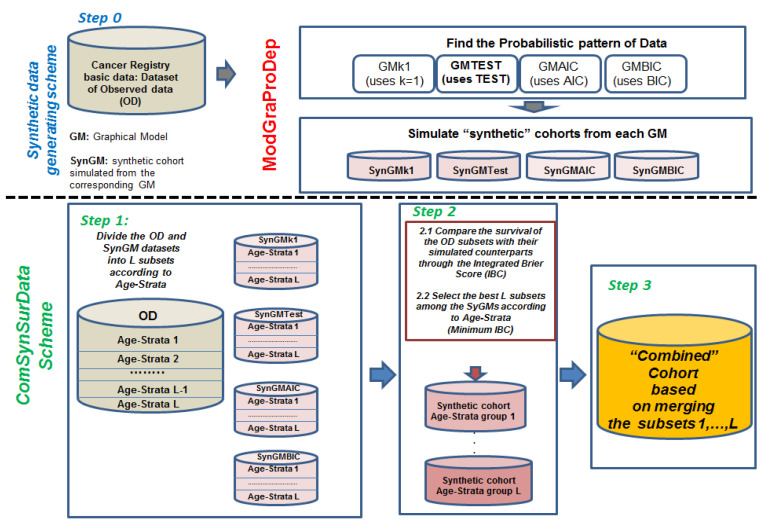
Scheme of procedure for generating combined cohort by using best synthetic cohort for each of the considered L age-stratum groups. Synthetic cohorts generated according to ModGraProDep.

**Figure 2 ijerph-19-03605-f002:**
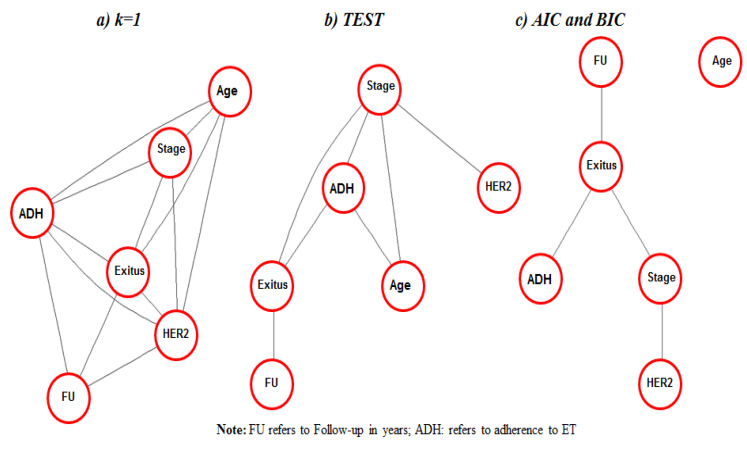
Undirected acyclic graphs generated from fitting the best graphical models to observed data (N = 1243) using different criterions: (**a**) GMK1 model: k-penalty factor of penalized log-likelihood set to 1; (**b**) GMTEST: testing for statistical significance of arches; (**c**) GMAIC: Akaike information criterion (BIC) and GMBIC: Bayesian information criterion (BIC).

**Figure 3 ijerph-19-03605-f003:**
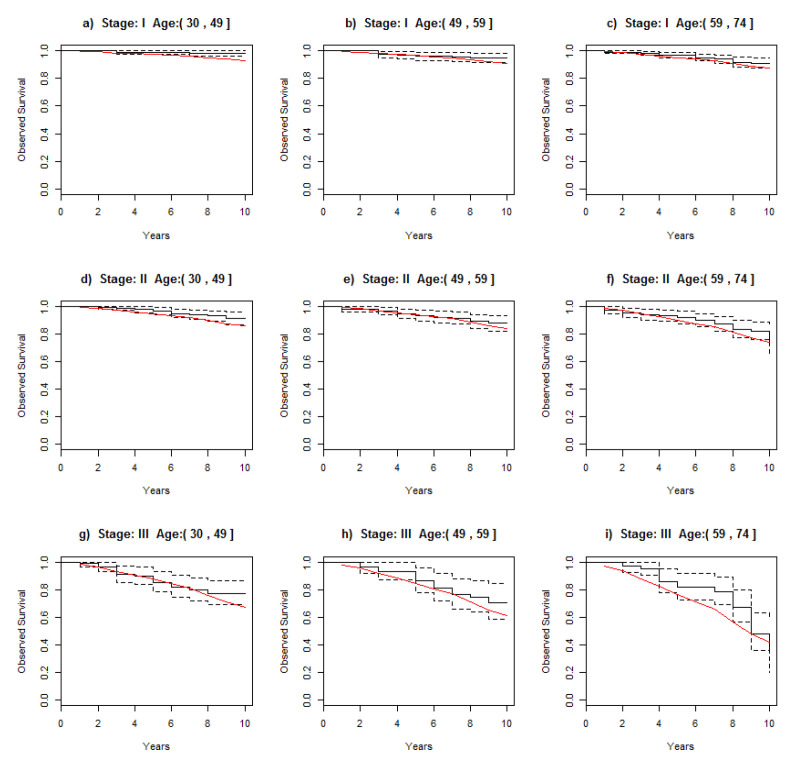
Comparison of 95% credible interval of observed survival derived from original cohort (black) and median survival (red) of combined cohort across stages at diagnosis and stratified by age group.

**Figure 4 ijerph-19-03605-f004:**
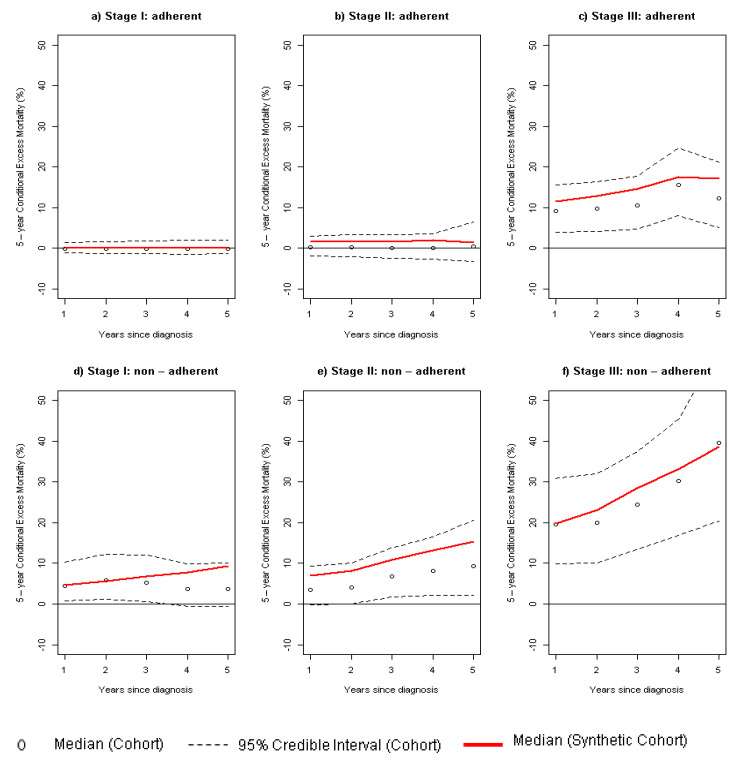
Comparison of 5-year conditional excess mortality (in percentage) between original cohort (black) and combined cohort (red) across stage at diagnosis and stratifying by adherence to endocrine therapy: (**a**,**d**) Stage I; (**b**,**e**) Stage II; (**c**,**f**) Stage III.

**Figure 5 ijerph-19-03605-f005:**
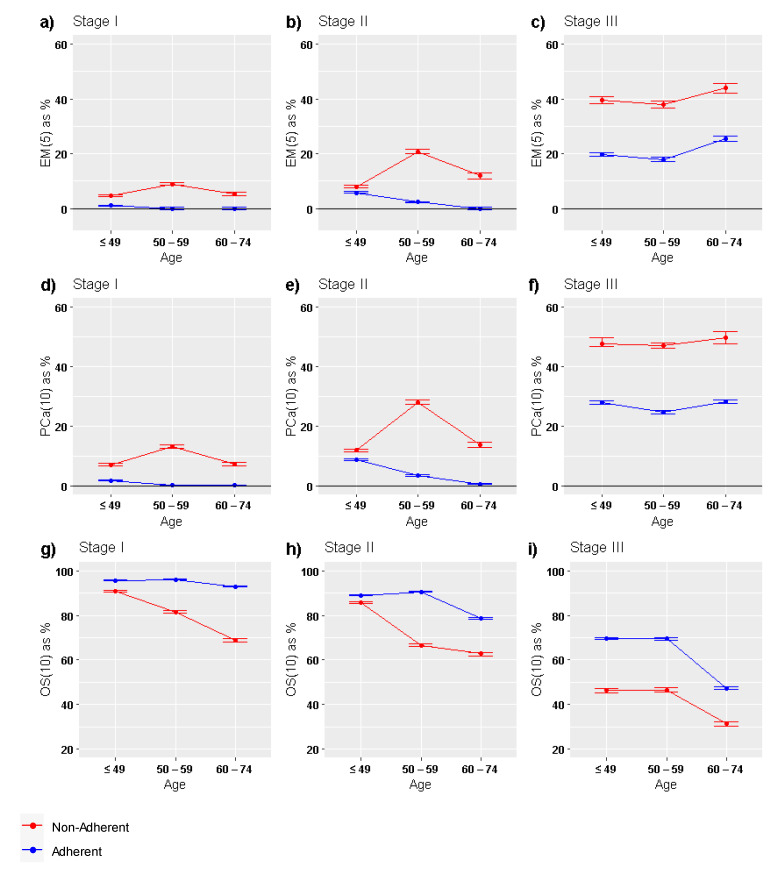
Graphical comparison of main population-based survival indicators between adherent and nonadherent BC patients across age groups: (**a**–**c**) EM(5); (**d**–**f**) PCa(10); (**g**–**i**) OS(10). EM(5): 5-year conditional excess mortality (EM) at *T* = 5 years after cancer diagnosis; PCa(10): crude probability of death due to cancer at *T* = 10 years; OS(10): observed survival at 10 years after cancer diagnosis.

**Table 1 ijerph-19-03605-t001:** Characteristics of patients diagnosed with breast cancer from 2005 to 2009 in Girona and Tarragona. Of the 2297 BC patients, complete data for endocrine treatment (*ET*) were available for 1243 in 2007–2009. Imputation of adherence through ModGraProDep was performed for the remaining 1054 BC patients.

		HER2− (N = 1736; 75.6%)	HER2+ (561; 24.4%)	Total (N = 2297; 100%)
**Registry**	Girona	876 (50.5%)	301 (53.6%)	1176 (51.2%)
	Tarragona	860 (49.5%)	260 (46.4%)	1121 (48.8%)
**Age**	Mean (SD)	55.6 (10.6)	54.3 (10.7)	55.3 (10.6%)
	≤49 years	556 (32.0%)	196 (35.0%)	751 (32.7%)
	50–59 years	502 (28.9%)	178 (31.7%)	680 (29.6%)
	60–74 years	678 (39.1%)	187 (33.3%)	866 (37.7%)
**Stage at diagnosis**	I	769 (44.3%)	195 (34.7%)	997 (43.4%)
	II	641 (36.9%)	257 (45.9%)	898 (39.1%)
	III	326 (18.8%)	109 (19.5%)	402 (17.5%)
**Deceased (%)**		11.9	10.9	11.7
**Follow-up in years, mean (SD)**		9.2 (1.7)	9.3 (1.5)	9.2 (1.6)
* **Adherence to ET** *	No: ≤80%	234 (13.4%; 24.9% ^b^)	75 (13.5%; 24.8% ^b^)	309 (13.5%; 24.9% ^b^)
	Yes: >80%	706 (40.7%; 75.1% ^b^)	228 (40.6%; 75.2% ^b^)	934 (40.6%; 75.1% ^b^)
	Total ^a^	940 (54.1%; 100.0% ^b^)	303 (54.0%; 100.0% ^b^)	1243 (54.1%; 100.0 ^b^)
	Missing ^c^	796 (45.9%; - )	258 (45.9%; - )	1054 (45.9%; - )
* **Distribution of BC Cases in Cohort after Imputation of Adherence to ET when Missing** *				
	* **Adherence to ET** *	* **HER2− (N = 1736; 75.6%)** *	* **HER2+ (N = 561; 24.4%)** *	* **Total (N = 2297; 100%)** *
**GMK1 ^d^**	No: ≤80%	426 (24.5%)	126 (22.4%)	552 (24.0%)
	Yes: >80%	1310 (74.5%)	435 (77.6%)	1745 (76.0%)
	Total	1736 (100.0%)	561(100.0%)	2297 (100.0%)
**GMAIC ^e^**	No: ≤80%	420 (24.2%)	120 (21.4%)	540 (23.5%)
	Yes: >80%	1316 (74.8%)	441 (78.6%)	1745 (76.5%)
	Total	1736 (100.0%)	561(100.0%)	2297 (100.0%)
**GMBIC ^f^**	No: ≤80%	420 (24.2%)	120 (21.4%)	540 (23.5%)
	Yes: >80%	1316 (74.8%)	441 (78.6%)	1745 (76.5%)
	Total	1736 (100.0%)	561(100.0%)	2297 (100.0%)
**GMTEST ^g^**	No: ≤80%	424 (24.4%)	121 (21.5%)	545 (23.7%)
	Yes: >80%	1312 (74.6%)	440 (78.5%)	1752 (76.3%)
	Total	1736 (100.0%)	561(100.0%)	2297 (100.0%)

^a^ Cases with available information on endocrine therapy in 2007–2009, N = 1243; ^b^ percentage with respect to ^a^; ^c^ cases with no available information on endocrine therapy; ^d–g^ distribution of cases according to adherence, imputing adherence status in BC cases with missing information by applying ModGraProDep models.

**Table 2 ijerph-19-03605-t002:** Integrated Brier score at up to 10 years of follow-up by age and stage, comparing the cohort’s absolute survival with the absolute survival estimated using each one of the synthetic datasets derived from the Graphical Models (in bold: minimal integrated Brier score for each age group according to stage of breast cancer at diagnosis).

	Synthetic Dataset			
	*Derived* *from* *GMk1*	*Derived* *from* *GMTest*	*Derived* *from* *GMAIC*	*Derived* *from* *GMBIC*
** *Stage I* **				
*≤49 years*	0.0149	0.0146	0.0142	0.0143
*50–59 years*	0.0432	0.0433	0.0433	0.0433
*60–74 years*	0.0471	0.0485	0.0482	0.0482
** *Stage II* **				
*≤49 years*	0.0484	0.0486	0.0485	0.0485
*50–59 years*	0.0703	0.0707	0.0706	0.0706
*60–74 years*	0.0943	0.0986	0.0982	0.0984
** *Stage III* **				
*≤49 years*	0.1183	0.1188	0.1185	0.1185
*50–59 years*	0.1437	0.1431	0.1427	0.1426
*60–74 years*	0.1722	0.2020	0.1960	0.1965

**Table 3 ijerph-19-03605-t003:** Survival indicators derived from synthetic cohort comparing breast cancer patients adherent vs. nonadherent to endocrine therapy across age groups and stage at diagnosis.

		OS(5) (%)	OS(10) (%)	PCa(10) (%)	POC(10) (%)	EM(5) (%)
Adherent	*N **	Me(95% CI)	Me(95% CI)	Me(95% CI)	Me(95% CI)	Me(95% CI)
Stage I						
≤49 years	72,817	98.3 (98.2; 98.4)	95.7 (95.5; 95.9)	1.8 (1.6; 1.9)	2.5 (2.5; 2.5)	1.1 (1.0; 1.3)
50–59 years	92,526	98.4 (98.3; 98.5)	96.0 (95.9; 96.2)	0.2 (0.1; 0.3)	3.8 (3.6; 3.9)	0.0 (−0.1; 0.1)
60–74 years	167,001	97.3 (97.2; 97.3)	92.9 (92.7; 93.1)	0.2 (0.1; 0.3)	6.9 (6.7; 7.1)	0.0 (−0.1; 0.1)
Stage II						
≤49 years	98,722	95.9 (95.7; 96.1)	89.0 (88.8; 89.3)	8.7 (8.4; 8.9)	2.3 (2.0; 2.6)	5.8 (5.5; 6.2)
50–59 years	92,612	96.5 (96.4; 96.6)	90.6 (90.4; 90.9)	3.4 (3.1; 3.6)	6.0 (5.9; 6.1)	2.3 (2.1; 2.6)
60–74 years	92,919	91.9 (91.7; 92.1)	78.6 (78.3; 79.0)	0.6 (0.4; 0.9)	20.8 (20.3; 21.1)	0.0 (−0.4; 0.4)
Stage III						
≤49 years	40,968	87.8 (87.5; 88.1)	69.6 (69.1; 70.1)	27.9 (27.3; 28.4)	2.5 (2.5; 2.6)	19.7 (19.1; 20.2)
50–59 years	36,659	88.0 (87.7; 88.3)	69.5 (68.9; 70.1)	24.8 (24.2; 25.4)	5.7 (5.7; 5.8)	17.9 (17.3; 18.6)
60–74 years	41,335	78.8 (78.4; 79.2)	47.3 (46.7; 47.9)	28.1 (27.5; 28.8)	24.6 (24.5; 24.7)	25.4 (24.5; 26.3)
**Overall**	735,559	94.5 (94.4; 94.6)	85.7 (85.6; 85.8)	0.9 (0.8; 1.0)	13.5 (13.3; 13.6)	0.5 (0.4; 0.7)
Nonadherent						
Stage I						
≤49 years	34,356	96.5 (96.3; 96.7)	90.8 (90.4; 91.2)	7.1 (6.7; 7.5)	2.2 (2.2; 2.2)	4.6 (4.3; 5.0)
50–59 years	29,888	92.8 (92.5; 93.1)	81.5 (80.9; 82.0)	13.1 (12.6; 13.7)	5.4 (5.4; 5.4)	9.0 (8.5; 9.6)
60–74 years	33,313	87.7 (87.3; 88.0)	68.8 (68.2; 69.5)	7.4 (6.8; 8.0)	23.9 (23.2; 24.1)	5.4 (4.6; 6.1)
Stage II						
≤49 years	49,897	94.7 (94.5; 94.9)	85.9 (85.5; 86.3)	11.9 (11.5; 12.3)	2.1 (2.1; 2.1)	8.0 (7.6; 8.4)
50–59 years	28,269	87.2 (86.8; 87.6)	66.6 (65.9; 67.4)	28.0 (27.3; 28.7)	5.3 (5.3; 5.4)	20.8 (20.1; 21.6)
60–74 years	26,084	85.6 (85.2; 86.0)	62.7 (61.9; 63.4)	13.9 (13.0; 14.7)	23.5 (23.4; 23.6)	11.9 (10.9; 12.9)
Stage III						
≤49 years	16,468	77.6 (77.0; 78.2)	46.3 (45.3; 47.3)	49.8 (48.9; 50.7)	3.9 (3.8; 4.0)	39.5 (38.3; 40.7)
50–59 years	14,953	78.0 (77.3; 78.6)	46.5 (45.5; 47.5)	47.0 (46.1; 47.9)	6.5 (6.4; 6.6)	38.0 (36.8; 39.3)
60–74 years	13,020	72.2 (71.4; 72.9)	31.3 (30.2; 32.3)	41.1 (40.1; 42.0)	27.7 (27.4; 28)	43.9 (42.1; 45.7)
**Overall**	246,248	88.5 (88.4; 88.6)	70.5 (70.3; 70.8)	19.6 (19.4; 19.9)	9.8 (9.6; 9.9)	14.5 (14.3; 14.8)
**Overall difference** **Adherent vs. nonadherent ****		6.0	15.2	−18.7	3.7	−14.0

**Note:** survival indicators expressed in percentage; **N *:** number of patients in the combined synthetic cohort; **Me:** Median; **95 CI:** 95% Credible Interval; **OS(T):** observed survival at T = 5 and T = 10 years after cancer diagnosis; **PCa(10)**: crude probability of death due to cancer at T = 10 years; **POC(10):** crude probability of death due to other causes at T = 10 years; **EM(T):** 5-year conditional excess mortality at T years after cancer diagnosis; **: difference in the median estimate Adherent minus nonadherent.

## Data Availability

Data supporting reported results can be requested to the corresponding author.

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
