# Peer review of "No Excess Mortality up to 10 Years in Early Stages of Breast Cancer in Women Adherent to Oral Endocrine Therapy: A Probabilistic Graphical Modeling Approach"

_ijerph, 2022, doi:10.3390/ijerph19063605_

Round 1
Reviewer 1 Report
I read this article with great interest. I think this article is well designed and performed. The method is solid, and the conclusion is reliable. Paper accepted.
The revised manuscript deserves attention since this is the first study to show that adherence to ET has a major impact on BC survival among HR+ patients with early stage breast cancer: no excess risk of death up to 10 years after BC in women diagnosed beyond 49 years of age. This result sheds light into the cure of BC for this group of patients.
So I believe that this manuscript is suitable for publication in your esteemed Journal.
This manuscript has two main limitations:
- In assessing the “statistical cure” of BC: the follow- up cannot go beyond 10 years.
- The simulated cohorts were based on the observed data provided by the original cohort
So I believe that the availability of external data provided by other cancer registries with similar information would be useful for an additional validation of the results and reproducibility.
Overall, this manuscript is written exceptionally well. The use of English language is very good, and the sections follow clearly.
Author Response
Reviewer 1 Comments and suggestions:
“I read this article with great interest. I think this article is well designed and performed. The method is solid, and the conclusion is reliable. Paper accepted.
The revised manuscript deserves attention since this is the first study to show that adherence to ET has a major impact on BC survival among HR+ patients with early stage breast cancer: no excess risk of death up to 10 years after BC in women diagnosed beyond 49 years of age. This result sheds light into the cure of BC for this group of patients.
So I believe that this manuscript is suitable for publication in your esteemed Journal.
This manuscript has two main limitations:
- In assessing the “statistical cure” of BC: the follow- up cannot go beyond 10 years.
- The simulated cohorts were based on the observed data provided by the original cohort
So I believe that the availability of external data provided by other cancer registries with similar information would be useful for an additional validation of the results and reproducibility.
Overall, this manuscript is written exceptionally well. The use of English language is very good, and the sections follow clearly.”
Response to reviewer 1 comments:
We thank the reviewer for his/her positive comments on out manuscript. The reviewer noted that the manuscript deserves attention since the results presented shed light into the cure of BC for this group of patients. We believe that this message is crucial for disseminating the clinical implications of the results presented here. Our aim was in this line.
The reviewer also noted two major limitations: (1) related with cancer “cure” assessment and related with (2) the availability of external data to validate our results in other studies/regions. These limitations were noted in the original version of the paper, and these can be found at the Discussion section third paragraph “Our study has …. useful for an additional validation of the results and reproducibility”.
Finally, we revised the English language of the paper by a native speaker.

Reviewer 2 Report
The work is very interesting and concurrently can improve the effectiveness of treatment and thereby improve survival od women with breast cancer. It is also needed to emphasize the originality of the used study method, a way of making the study and presentation of its results. The number of people involved in the study also deserves to be emphasized. I congratulate the Authors and fully recommend this article for the publication.
Author Response
Reviewer 2 Comments and suggestions:
“The work is very interesting and concurrently can improve the effectiveness of treatment and thereby improve survival of women with breast cancer. It is also needed to emphasize the originality of the used study method, a way of making the study and presentation of its results. The number of people involved in the study also deserves to be emphasized. I congratulate the Authors and fully recommend this article for the publication.”
Response to reviewer 2 comments:
We thank the reviewer for his/her positive comments on our manuscript. As noted by the reviewer, “The results presented here can improve effectiveness of treatment and thereby improve survival of women with BC”. We believe that this message has clinical implications and the results presented here are with this aim.

Reviewer 3 Report
This work uses a new approach and a new tool with potential utility for the validation of survival indicators. Specifically, it has been useful in the population study of breast cancer treated with hormone therapy. In my opinion it has the necessary quality to be published.
Author Response
Reviewer 3 Comments and suggestions:
“This work uses a new approach and a new tool with potential utility for the validation of survival indicators. Specifically, it has been useful in the population study of breast cancer treated with hormone therapy. In my opinion it has the necessary quality to be published.”
Response to reviewer 3 comments:
We thank the reviewer for his/her positive comments on our manuscript. As noted by the reviewer, the method used in this paper has a potential validation of population-based survival indicators. Our aim was in this line and in showing the clinical implications of BC patient’s adherence to endocrine therapy when prescribed.

Reviewer 4 Report
The authors of the manuscript presented the potential use of probabilistic graphical modeling for generating reliable synthetic datasets for validating population-based survival indicators when sample size is an issue.
My comments:
The method and the results are described in a very mathematical language, of course it is understandable. But I think International Journal of Environmental Research and Public Health covers e.g. Public Health, Environmental Health and Global Health Research and it would be good to add a graphic Figures the results obtained to the "Chapter Results", which will make the work more accessible to clinicians. It will also allow a better understanding of the application of Graphical Models through ModGraProDep in Oncology and Public Health.
My minor remark:
Figure 2 is illegible - please correct it.
Figure 3 is out of focus - please correct it.
Author Response
Reviewer 4 Comments and suggestions:
“The authors of the manuscript presented the potential use of probabilistic graphical modeling for generating reliable synthetic datasets for validating population-based survival indicators when sample size is an issue.
My comments:
The method and the results are described in a very mathematical language, of course it is understandable. But I think International Journal of Environmental Research and Public Health covers e.g. Public Health, Environmental Health and Global Health Research and it would be good to add a graphic Figures the results obtained to the "Chapter Results", which will make the work more accessible to clinicians. It will also allow a better understanding of the application of Graphical Models through ModGraProDep in Oncology and Public Health.
My minor remark:
Figure 2 is illegible - please correct it.
Figure 3 is out of focus - please correct it.”
Response to reviewer 4
Response to “My comments”:
We would like to thank the reviewer for his/her thoughtful comments and efforts towards improving our manuscript. As noted by the author, the paper shows the (i) application of Graphical Modeling, and (ii) how to use survival indicators derived from a large sample synthetic dataset derived from the original data fitted to several graphical models (i).
In this line, we presented mathematical details of the methodology used in this paper since it extends the Graphical modeling approach presented in ModGraProDep (Vilardell et al 2020). The paper uses a resampling approach, ComSynSurData, for generating an “optimal” simulated dataset from which one can derive reliable survival indicators. We believe that a detailed presentation of these methods in our study shows how to implement and adapt the method for its use in tumors other than BC.
However, the indicators derived from this modeling must be understood and accessible to clinicians. Table 3 presents the indicators for clinical use, but their interpretation must be improved through a graphical comparison according to adherent and non-adherent groups and across age-groups. For this purpose, in the revised version of the paper we also included the new Figure 5, (see page 13) which shows the differences in the main population-based survival indicators when comparing adherent versus non-adherent patients. The last paragraph in page 12 comments this Figure “Figure 5 shows the comparison…”. In addition, Figure 5 also shows that the differences found between adherent and non-adherent patients have a significant impact but of different magnitude depending on age, stage and adherence to ET. This is also included in the Discussion section, page 14, end of 3rd paragraph: “Moreover, ….”.
Response to “Myminor remark Figure 2 is illegible - please correct it.”
We thank the reviewer for pointing this out. We have corrected the figure accordingly and also increased its quality for publication. (see page 8)
Response to “Myminor remark: Figure 3 is out of focus - please correct it”
We thank the reviewer for pointing this out. We have corrected the figure accordingly and also increased its quality for publication. (See page 10).

This manuscript is a resubmission of an earlier submission. The following is a list of the peer review reports and author responses from that submission.
Round 1
Reviewer 1 Report
The manuscript entitled “No excess mortality up to 10 years in early stages of breast cancer in women adherent to oral endocrine therapy: a probabilistic graphical modeling approach” by Ramon Clèries et al, aims to assess 10-year excess mortality (EM) in hormone receptor positive breast cancer (BC) patients’ which are adhered to oral endocrine therapy (ET) by using population-based cancer registry data from Spain. The authors have developed ComSynSurData, which is a probabilistic graphical modeling method for generating a large cohort of synthetic data that mimics the survival patterns of the original cohort.
General comments:
In the present study authors have observed and the study concludes that there is clue for developing interventions to promote adherence to ET in patients with hormone receptor-positive BC and in turn improving cancer survival, however in continuation to this, I wonder whether this method would be valid for other cancers too or this is probabilistic approach for the specific cancer only.